# IFormerFusion: Cross-Domain Frequency Information Learning for Infrared and Visible Image Fusion Based on the Inception Transformer

Zhang Xiong 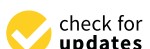, Xiaohui Zhang *, Qingping Hu and Hongwei Han

Department of Weapon Engineering, Naval University of Engineering, Wuhan 430030, China
* Correspondence: zhangxiaohui505@163.com

**Abstract:** The current deep learning-based image fusion methods can not sufficiently learn the features of images in a wide frequency range. Therefore, we proposed IFormerFusion, which is based on the Inception Transformer and cross-domain frequency fusion. To learn features from high- and low-frequency information, we designed the IFormer mixer, which splits the input features through the channel dimension and feeds them into parallel paths for high- and low-frequency mixers to achieve linear computational complexity. The high-frequency mixer adopts a convolution and a max-pooling path, while the low-frequency mixer adopts a criss-cross attention path. Considering that the high-frequency information relates to the texture detail, we designed a cross-domain frequency fusion strategy, which trades high-frequency information between the source images. This structure can sufficiently integrate complementary features and strengthen the capability of texture retaining. Experiments on the TNO, OSU, and Road Scene datasets demonstrate that IFormerFusion outperforms other methods in object and subject evaluations.

**Keywords:** image fusion; transformer; inception transformer; infrared image; visible image

## 1. Introduction

The image fusion technique aims to fuse images captured by different sensors to generate a fused image with better human visual effects and scene representation. As a result of thermal-based imaging, the infrared sensor can work in low-illuminance environments and all weather conditions and generate infrared images to emphasize prominent targets. The infrared image contains rich, low-frequency information, but it lacks texture detail. The visible image contains a rich texture and detailed spatial features, but it is susceptible to influence by illuminance and weather conditions. Due to the good complementarity of infrared and visible images, the fused image can provide high-quality results for person re-identification [1], object tracking [2], remote sensing [3,4], and salient object detection [5].

The image fusion methods can be categorized into traditional methods and deep learning-based methods. Traditional methods usually consist of three parts, a manual representation model to extract features, fusion strategies to the feature maps or weight maps, and an inverse feature extractor to reconstruct images. The traditional methods can be categorized into spatial domain [6] and frequency domain methods [7–9]. The spatial domain methods are usually simple to infer, but have poor effects on edge preserving. The frequency domain methods usually adopt domain transform operation and fuse procedures, which are more complex [10]. Although traditional fusion methods can achieve good fusion results, the manual design of transformation algorithms and fusion strategies can result in higher computational complexity, limiting the fusion performance. Conversely, deep learning-based models can extract features and generate high-performance fusion images without the need for a complex manual design. Deep learning-based methods can be categorized into four types: convolution neural network(CNN)-based fusion methods [11,12], auto-encoder-based fusion methods [13,14], generative adversarial network(GAN)-based

fusion methods [15,16], and transformer-based fusion methods [17,18]. Although the above methods can generate fused images with a good performance, some issues still exist in order for them to be improved. CNN-based, auto-encoder-based, and GAN-based fusion methods generally use convolutional layers to extract the features. Thus, these methods cannot establish long-range dependencies with the limitation of the perceptive field of convolutional layers. The vision transformer develops attention mechanisms to build a long-range relationship between the image patches. However, there are still some drawbacks. Firstly, the existing methods fail to learn information in a wide frequency range, which is important for infrared and visible image fusion tasks. Secondly, the existing methods a lack consideration for the relationship between the frequency information of the source images.

Above all, we proposed IFormerFusion based on the Inception Transformer and cross-domain frequency fusion. On the one hand, we designed the IFormer mixer based on the Inception Transformer, which adopts convolution/max-pooling paths to process high-frequency information and a criss-cross attention path to process low-frequency information. The IFormer mixer was designed to capture high-frequency information to Transformers structure and learn information in a wide frequency range. On the other hand, cross-domain frequency fusion can trade high-frequency information between the source images to guide the model to retain more high-frequency information. IFormerFusion has four parts: feature extraction, cross-domain frequency fusion, feature reconstruction, and fused image reconstruction. The feature extractor can effectively extract comprehensive features in a wide frequency range. The cross-domain frequency fusion can learn features in a wide frequency range and trade high-frequency information between the source images to strengthen the texture-retaining capability of the model. The fusion results are concatenated and fed to the Inception Transformer-based feature reconstruction part to reconstruct deep features. Finally, a CNN-based fused image reconstruction part is utilized to reconstruct the images. Above all, the main contributions of this work can be summarized as follows:

- We propose an infrared and visible image fusion method, IFormerFusion, which can efficiently learn features from source images in a wide frequency range. IFormerfusion can sufficiently retain texture details and maintain the structure of the source images.
- We designed the IFormer mixer, which consists of convolution/max-pooling paths and a criss-cross attention path. The convolution/max-pooling can learn high-frequency information, while the criss-cross attention path can learn low-frequency information.
- The cross-domain frequency fusion can trade high-frequency information between the source images to sufficiently learn comprehensive features and strengthen the capability to retain texture.
- Experiments conducted using the TNO, OSU, and Road Scene datasets show that IFormerFusion obtains better results in both visual quality evaluation and quantatively evaluation.

## 2. Related Works

### 2.1. Vision Transformer

The transformer, which was designed for natural language processing (NLP), has achieved notable success in a broad range of computer vision tasks, such as image classification [19,20], semantic segmentation [21,22], and object detection [23–28]. In 2020, Dosovitskiy et al. proposed the vision transformer, which splits an image into a sequence of flattened $16 \times 16$ patches and regards them as words in pieces of text [29]. Then, the vision transformer will embed the patches linearly and adopt a self-attention mechanism, which also brings an increased computational complexity quadratic to the image size, which is the bottleneck of the application in pixel-level tasks (such as image fusion) that require dense prediction. In 2021, Liu et al. proposed the Swin Transformer, in which self-attention computation is limited in the local window and shifted window partitioning in successive blocks [21]. Thus, the Swin Transformer has a linear complexity corresponding to the image size [21]. In 2022, Si et al. noticed that it is incompetent at learning information in a wide

frequency range and proposed the Inception Transformer, which can sufficiently learn comprehensive features with both high- and low-frequency information [30]. The input images are split into three partitions and fed into the Inception mixer to learn features in a wide frequency range. Through the Inception mixer, the Inception Transformer has greater efficiency through a channel splitting mechanism to adopt parallel convolution/max-pooling paths and self-attention path to learn information within a wide frequency range [30].

### 2.2. Deep Learning-Based Image Fusion Methods

In the beginning, deep learning models were only utilized to extract features and generate weight maps for fusion [11]. The majority of fusion methods retain the structure of the traditional fusion framework. With more researchers designing networks and loss functions, CNN-based methods gradually differ from traditional frameworks. In 2021, Long et al. designed a parallel, aggregated, residual dense block consisting of a dense block path and a residual dense block path and the proposed RXDNFuse [12]. In 2019, Li et al. introduced an auto-encoder-based fusion network, DenseFuse [13]. The auto-encoder-based fusion methods can benefit from the high interpretability of the traditional fusion methods and the feature extraction ability of the CNN. Thus, many researchers focus on improving each part of the auto-encoder structure. In 2020, to fuse the extracted feature more effectively and reconstruct the feature maps, Li et al. proposed NestFuse, which adopts a nest-connected decoder network and the attention-based fusion strategy [31]. In 2021, Xu et al. proposed a learnable fusion strategy for the first time, which could quantitatively measure the classification significance of feature maps by using the back-propagation integral gradient of the classification results [32]. In 2022, Wang et al. proposed Res2Fusion, which adopts a multi-scale feature extraction strategy without down-sampling, and established long-distance feature dependency through nonlocal attention mechanisms [33].

Some researchers utilized the generative adversarial network(GAN) to fuse images in an implicit manner. In 2019, Ma et al. first introduced GAN to infrared and visible image fusion and proposed FusionGAN [15]. Subsequently, they proposed other GAN-based fusion methods (DDcGAN [34] in 2020 and GANMcC [16] in 2021). DDcGAN is utilized to fuse multi-resolution images. GANMcC is utilized to generate fused images with a balance between the gradient and intensity of the source images.

To further improve the fusion effects, some researchers replaced the CNN layers with transformer structures. In 2022, Liu et al. proposed MFST, which adopts a self-adaptive transformer fusion strategy [14]. In 2022, Rao et al. proposed TGFuse, which adopts a transformer-based generator [35]. In 2022, Wang et al. proposed SwinFuse, which adopts residual Swin Transformer blocks as the encoder network. Based on the Swin Transformer, in 2022, Ma et al. proposed a cross-domain long-range fusion method, which includes inter- and Swin Transformer-based cross-domain modules to extract and fuse deep features [18].

Different from the mentioned methods, we propose IFormerFusion, which sufficiently learns the features of the images in a wide frequency range through the IFormer mixer that consists of parallel convolution, max-pooling, and attention paths in a wide frequency range. Moreover, we designed a cross-domain frequency fusion strategy to sufficiently integrate complementary features and strengthen the ability to retain texture.

## 3. Methodology

### 3.1. Overall Framework

Let $I_1$ and $I_2 \in \mathbb{R}^{H \times W \times C}$ represent the input images of IFormerFusion, while C is the channel number, and H and W are the image sizes. As shown in Figure 1, IFormerFusion is constructed with four parts: feature extraction, cross-domain frequency fusion, feature reconstruction, and fused image reconstruction.

Feature Extraction: Firstly, the input images or features are first embedded by convolutional embedding layers, which extend the channel dimension of the input to the required

number. In this experiment, the required channel number is 60. The embedding result can be expressed as:

$$\boldsymbol{\phi} = Embed(\boldsymbol{I}), \tag{1}$$

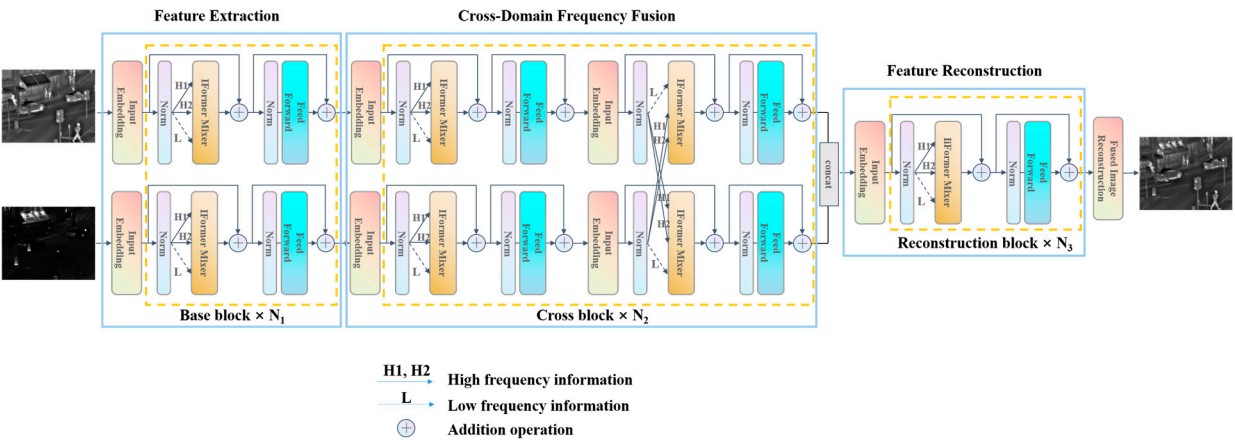

**Figure 1.** The framework of IFormerFusion.

We then design feature extraction blocks called base blocks. In each base block, the input feature will split into three partitions including two high-frequency partitions (H1 and H2) and a low-frequency partition (L) through the channel dimension and feed them into the IFormer mixer. Next, a feed-forward network (FFN) is deployed to refine the result of the IFormer mixer. Layer normalization (LN) is adopted before both the FFN and the split operation in the IFormer mixer, which is inferred specifically in the next section. Moreover, the residual connection is adopted in the mixer. The inference can be expressed as:

$$\boldsymbol{\phi}_{h1}, \boldsymbol{\phi}_{h2}, \boldsymbol{\phi}_l = Split(Norm(\boldsymbol{\phi})), \tag{2}$$

$$\boldsymbol{\phi}_M = \boldsymbol{\phi} + Mixer(\boldsymbol{\phi}_{h1}, \boldsymbol{\phi}_{h2}, \boldsymbol{\phi}_l), \tag{3}$$

$$\boldsymbol{\phi}_F = \boldsymbol{\phi}_M + FFN(Norm(\boldsymbol{\phi}_M)), \tag{4}$$

where $\boldsymbol{\phi}_{h1}$ and $\boldsymbol{\phi}_{h2}$ represent the high-frequency information, $\phi_l$ represents the low-frequency information, $\boldsymbol{\phi}_M$ represents the residual mixed results, and $\phi_F$ represents the extract features. Feature extraction consists of $N_1$ base blocks. In this experiment, $N_1$ is 3.

Cross-Domain Frequency Fusion: The cross-domain frequency fusion part is designed to learn features in a wide frequency range and fuse cross-domain frequency information. The cross-domain frequency fusion part consists of $N_2$ cross blocks. The base block is the same as the base block in feature extraction. However, in the cross block, the high-frequency information of two input features is exchanged. The embedding results of input features $\boldsymbol{I_1}$ and $\boldsymbol{I_2}$ can be expressed as:

$$\boldsymbol{\phi_1} = Embed(\boldsymbol{I_1}), \tag{5}$$

$$\boldsymbol{\phi_2} = Embed(\boldsymbol{I_2}), \tag{6}$$

The split partitions of $\boldsymbol{\phi_i}$ are $\boldsymbol{\phi}_{h_1}^i$, $\boldsymbol{\phi}_{h_2}^i$, and $\boldsymbol{\phi}_l^i$ ($i = 1, 2$), which can be expressed as:

$$\boldsymbol{\phi}_{h_1}^1, \boldsymbol{\phi}_{h_2}^1, \boldsymbol{\phi}_l^1 = Split(Norm(\boldsymbol{\phi_1})), \tag{7}$$

$$\boldsymbol{\phi}_{h_1}^2, \boldsymbol{\phi}_{h_2}^2, \boldsymbol{\phi}_l^2 = Split(Norm(\boldsymbol{\phi_2})), \tag{8}$$

The mixed results $\boldsymbol{\phi}_M^1$ and $\boldsymbol{\phi}_M^2$ can be expressed as:

$$\boldsymbol{\phi}_M^1 = \boldsymbol{\phi_1} + Mixer\left(\boldsymbol{\phi}_{h_2}^2, \boldsymbol{\phi}_{h_2}^2, \boldsymbol{\phi}_l^1\right), \tag{9}$$

$$\phi_M^2 = \phi_2 + Mixer\left(\phi_{h_1}^1, \phi_{h_2}^1, \phi_l^2\right), \tag{10}$$

The extract features $\phi_F^1$ and $\phi_F^2$ can be expressed as:

$$\phi_F^1 = \phi_M^1 + FFN\left(Norm\left(\phi_M^1\right)\right), \tag{11}$$

$$\phi_F^2 = \phi_M^2 + FFN\left(Norm\left(\phi_M^2\right)\right), \tag{12}$$

In this experiment, $N_2$ is 6. The results $\phi_F^1$ and $\phi_F^2$ are concatenated and fed to the feature reconstruction part. The concatenated result $\phi_F$ can be expressed as:

$$\phi_F = Concat\left(\phi_F^1, \phi_F^2\right), \tag{13}$$

Feature Reconstruction: The results of the Cross-Domain Frequency Fusion part are fed into the Feature Reconstruction part, which consists of $N_3$ reconstruction blocks. The reconstruction block is half of the base block, which has a single path to process the concatenated result. In this experiment, $N_3$ is 3.

Fused Image Reconstruction: Finally, a simple CNN-based part to reconstruct fused images is utilized to reconstruct fused images. The fused image reconstruction part consists of two convolutional layers.

### 3.2. IFormer Mixer

The architecture of the IFormer mixer is shown in Figure 2. After splitting the input feature into three partitions through the channel dimension, high-frequency and low-frequency mixers are adopted to learn the features in a wide frequency range. The high-frequency mixer has a max-pooling (MaxPool) path, which consists of a max pooling subsequently a linear layer [36], and a parallel convolution path, which consists of a linear subsequently a depthwise convolution (DwConv) layer [37]. The low-frequency mixer has a criss-cross attention path, which consists of average pooling, criss-cross attention (CC-Atten) [38], and up-sampling (UpSample). The detailed inference is as follows.

With the input $X \in \mathbb{R}^{H \times W \times C}$, $X$ can be divided into two parts through channel dimension, the high-frequency part, $X_h \in \mathbb{R}^{H \times W \times C_h}$, and the low-frequency part, $X_l \in \mathbb{R}^{H \times W \times C_l}$, where $C_h + C_l = C$. Then, $X_h$ and $X_l$ are assigned to the high-frequency mixer and low-frequency mixer, respectively. The details of the high- and low-frequency mixers follow.

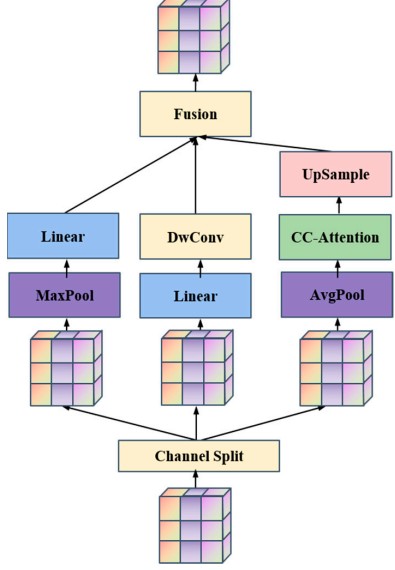

**Figure 2.** The architecture of the IFormer mixer.

High-frequency mixer: Considering the sharpness sensitivity of the maximum filter and the detail perception of the convolution operation, we propose two high-frequency paths to take advantage of the sharpness sensitivity of max-pooling and the detail perception capability of the convolution layers to learn high-frequency information. Firstly, the input $X_h$ is divided into $X_{h1} \in \mathbb{R}^{H \times W \times C_{h/2}}$ and $X_{h2} \in \mathbb{R}^{H \times W \times C_{h/2}}$. $X_{h1}$ is fed into the max pooling path. $X_{h2}$ is fed into the parallel convolution path. The outputs of the high-frequency mixer $Y_{h1}$ and $Y_{h2}$ can be expressed as:

$$Y_{h1} = Linear(MaxPool(X_{h1})), \tag{14}$$

$$Y_{h2} = DwConv(Linear(X_{h2})), \tag{15}$$

Low-frequency mixer: Considering the strong capability of the attention mechanism for learning global representation, we use criss-cross attention [38] to establish long-range dependency to learn low-frequency information. However, in image fusion tasks, dense prediction can bring great computation complexity with the large resolution of feature maps in the low-frequency mixer. Therefore, we utilize an average-pooling operation to reduce the scale of $X_l$ before the criss-cross attention operation and an up-sample layer to restore the original scale. In this experiment, the kernel size and stride for the average pooling are 4, and the size of up-sample layers is also 4. This branch can be defined as

$$Y_l = Upsample(CC(AvgPool(X_l))), \tag{16}$$

where $Y_l$ is the output of the low-frequency mixer, and CC represents criss-cross attention.

Finally, the outputs $Y_{h1}$, $Y_{h2}$, and $Y_l$ are concatenated through the channel dimension and fused to obtain $Y_c$:

$$Y_c = Fusion(Concat(Y_{h1}, Y_{h2}, Y_l)), \tag{17}$$

### 3.3. Loss Function

Three loss functions are adopted for the IFormerFusion in the training phase, which is explained as follows.

The structural similarity (SSIM) loss $L_{SSIM}$ can be expressed as:

$$L_{SSIM} = \alpha(1 - SSIM(I_F, I_1) + \beta(1 - SSIM(I_F, I_2)), \tag{18}$$

where $\alpha = \beta = 0.5$ in this experiment and $SSIM(\cdot)$ represents the philosophy of structural similarity [39].

Inspired by IFCNN [40] and SwinFusion [18], we deploy the intensity loss to supervise the model to capture potential intensity information. The intensity loss $L_{Int}$ can be expressed as:

$$L_{Int} = \frac{1}{HW} \|I_F - max(I_1, I_2)\|_1, \tag{19}$$

where $\| \cdot \|1$ represents the l1-norm, and $max(\cdot)$ represents the chosen max value. $H$ and $W$ represent the image sizes.

We deploy texture loss [18] to evaluate the texture details, which can be extracted by the maximum function. Thus, the texture loss $L_{Text}$ can be expressed as:

$$L_{Text} = \frac{1}{HW} \cdot \|\nabla I_F - max(|\nabla I_1|, |\nabla I_2|)\|_1, \tag{20}$$

where $\nabla$ represents the Sobel gradient operator, $|\cdot|$ represents for the absolute operation, $max(\cdot)$ represents to choose max value, and $\| \cdot \|1$ represents the l1-norm.

Finally, the total loss function $L_{total}$ is a weighted sum of all loss functions, which can be expressed as:

$$L_{total} = \lambda_1 L_{SSIM} + \lambda_2 L_{Int} + \lambda_3 L_{Text}, \tag{21}$$

where $\lambda_1$, $\lambda_2$, and $\lambda_3$ are weighted to balance each loss.

## 4. Experimental Results and Analysis

In this section, IFormerFusion is compared with eight advanced fusion methods with images selected from three public datasets. Subsequently, we conducted quantitative and qualitative comparisons with nine advanced methods: DenseFuse [13], GANMcC [16], IFCNN [40], NestFuse [31], Res2Fusion [33], RFN-Nest [41], SwinFuse [42], SwinFusion [18], and U2Fusion [43]. Finally, a computational complexity analysis is conducted.

### 4.1. Experiment Setup

The MSRS dataset [44] is selected for training. A total of nine deep learning methods are selected to compare the methods, i.e., DenseFuse, GANMcC IFCNN, NestFuse, Res2Fusion, RFN-Nest, SwinFuse, SwinFusion, and U2Fusion. The TNO image fusion dataset [45], OSU Color-Thermal dataset [46], and RoadScene dataset [43] are selected to evaluate the above methods.

All the experiments in this paper are conducted on Intel(R) Xeon(R) Silver 4210 CPU and NVIDIA GeForce RTX 3090 GPU. The PyTorch program is used. The batch size is eight. The images are randomly cropped to $128 \times 128$ patches and normalized to [0, 1]. The Adam optimizer is used. The model is trained for 200 epochs. The initial learning rate is 0.001 and it decays to half this in epochs 20, 40, 80, 120, and 180.

### 4.2. Evaluation Metrics

A total of six metrics are selected for evaluation, including mutual information (MI) [47], fast mutual information (FMI) [48], the peak signal-to-noise ratio (PSNR), the structural similarity index measurement (SSIM) [39], visual information fidelity (VIF) [49], and $Q_{abf}$ [50]. These metrics measure the performance of the fusion method from different aspects. Suppose the infrared image, the visible image, and the fused image are I, V, and F, respectively. Their detailed definitions are described as follows:

The mutual information metric is a quality index that measures the amount of information transferred from the source images to the fused image. mutual information is a fundamental concept in information theory and measures the dependence of two random variables. The definition of the mutual information metric can be expressed as:

$$MI = MI(I, F) + MI(V, F),\tag{22}$$

where $MI(I, F)$ and $MI(V, F)$ are the amounts of information transferred from the infrared images and visible images to the fused image, respectively. The MI between two random variables can be calculated by the Kullback–Leibler approach, which can be expressed as:

$$MI(\boldsymbol{X}, \boldsymbol{F}) = \sum_{x,f} P_{\boldsymbol{X},\boldsymbol{F}}(x, f) log \frac{P_{\boldsymbol{X},\boldsymbol{F}}(x, f)}{P_{\boldsymbol{X}}(x) P_{\boldsymbol{F}}(f)}\tag{23}$$

where $P_{\boldsymbol{X},\boldsymbol{F}}(x, f)$ is the joint histogram of the source image $x$ and the fused image $\boldsymbol{F}$; $P_{\boldsymbol{X}}(x)$ and $P_{\boldsymbol{F}}(f)$ are the marginal histograms of the source image $\boldsymbol{X}$ and the fused image $\boldsymbol{F}$, respectively.

The fast mutual information metric calculates the regional mutual information between the corresponding windows in the fused image and the source images [48]. The mutual information $Info(\boldsymbol{I}, \boldsymbol{F})$ and $Info(\boldsymbol{V}, \boldsymbol{F})$ can be expressed as:

$$Info(\boldsymbol{X}, \boldsymbol{F}) = \frac{2}{n} \sum_{i=1}^{n} \frac{Info_i(\boldsymbol{X}, \boldsymbol{F})}{H_i(\boldsymbol{X}) + H_i(\boldsymbol{F})}\tag{24}$$

where $H_i(X)$ and $H_i(F)$ are the entropies of $X$ and $F$, and the mutual information $Info_i(X, F)$ can be expressed as:

$$Info_i(X, F) = \frac{H_i(X) + H_i(F)}{2} \tag{25}$$

The fast mutual information metric can be expressed as:

$$FMI(I, V, F) = \frac{1}{2}(Info(I, F) + Info(V, F)) \tag{26}$$

The peak signal-to-noise ratio metric is the ratio of the maximum possible power of a signal to the destructive noise power, which affects its representation accuracy. The PSNR of the fused image and source images can be expressed as:

$$PSNR = 10log\left(\frac{max(F)}{MSE(I, F)}\right) + 10log\left(\frac{max(F)}{MSE(V, F)}\right) \tag{27}$$

where $max(F)$ is the maximum value in the fused image, and the $MSE(I, F)$ and $MSE(V, F)$ are the mean-squared errors, which can be expressed as:

$$MSE(X, F) = \frac{1}{M * N} \sum_{i=0}^{M-1} \sum_{j=0}^{M-1} [X(i, j) - F(i, j)]^2 \tag{28}$$

where $X(i, j)$ and $F(i, j)$ are the value of $X$ and $F$ in row $i$ and column $j$, and $M$ and $N$ are the weight and height of the images, respectively.

The structural similarity index measurement metric calculates the similarity of the fused images and source images in terms of luminance, contrast, and structure [39]. The SSIM metric can be expressed as:

$$SSIM(X, F) = \frac{\left(2\mu_x\mu_f + C_1\right)(2\sigma_{xf} + C_2)}{\left(\mu_x^2 + \mu_f^2 + C_1\right)(\sigma_x^2 + \sigma_f^2 + C_2)} \tag{29}$$

where $\mu_x$ and $\mu_f$ are the average gradients of $X$ and $F$; $\sigma_x$ and $\sigma_f$ are the standard deviation of $X$ and $F$; $\sigma_{xf}$ is the correlation coefficient of $X$ and $F$; $C_1$, $C_2$, and $C_3$ are $(0.01 * L)^2$, $(0.03 * L)^2$, and $\frac{1}{2}(0.01 * L)^2$, respectively; $L$ the dynamic range of the pixel values, which is 255.

The visual information fidelity metric calculates image distortions including additive noise, blurs, and changes [49]. VIF is derived from the quantification of two types of mutual information: the mutual information between the input and the output of the HVS channel (described via a stationary white Gaussian noise model) when no distortion channel is presented (i.e., reference mutual information) and the mutual information between the input of the distortion channel and the output of the HVS channel for the test image [51]. The visual information fidelity can be expressed as:

$$VIF(X, F) = \frac{\sum_{k \in subbands} I(\vec{C}_{s,k}, \vec{F}_{s,k} | R_{S,k})}{\sum_{k \in subbands} I(\vec{C}_{s,k}, \vec{X}_{s,k} | R_{S,k})} \tag{30}$$

where the *subbands* are a collection of specific sub-bands; $I(\vec{C}_{s,k}, \vec{F}_{s,k} | R_{S,k})$ and $I(\vec{C}_{s,k}, \vec{F}_{s,k} | R_{S,k})$ are the information extracted from specific sub-bands of the source image and fused image; $\vec{C}_{s,k}$ is the random field from source images; $\vec{F}_{s,k}$ and $\vec{X}_{s,k}$ are the output of the HVS channel for the fused image and source images; $R_{S,k}$ is the model parameter of the sub-bands.

Thus, the visual information fidelity for fusion (VIFF) can be expressed as:

$$VIFF = \frac{1}{2}(VIF(\mathbf{V}, \mathbf{F}) + VIF(\mathbf{I}, \mathbf{F}))$$ (31)

The $Q_{abf}$ metric measures the similarity of the edge transferred from the source images to the fused image. $Q_{abf}$ can be expressed as:

$$Q_{abf} = \frac{\sum_{i=1}^{M} \sum_{j=1}^{N} (Q^{A,F}(i,j)w_A(i,j) + Q^{B,F}(i,j)w_B(i,j))}{\sum_{i=1}^{M} \sum_{j=1}^{N} (w_A(i,j) + w_B(i,j))}$$ (32)

where $w_X(i,j)$ is the weight matrix of source images and $Q^{X,F}(i,j)$ is the edge information transferred from the source image to the fused image. The $Q^{X,F}(i,j)$ can be expressed as:

$$Q^{X,F}(i,j) = Q_g^{X,F}(i,j)Q_a^{X,F}(i,j)$$ (33)

where $Q_g^{X,F}(i,j)$ and $Q_a^{X,F}(i,j)$ are the retaining edge intensity and direction in the pixel $(i,j)$, respectively.

*4.3. Results on the TNO Dataset*

Visual Quality Evaluation: Four pairs of images are selected to evaluate the visual effects of IFormerFusion and nine comparable methods, as shown in Figure 3. Some targets and details are highlighted with boxes to display the information that is worthy of attention. DenseFuse, NestFuse, RFN-Nest, and SwinFuse extract features through a well-designed encoder and adopt an additional attention-based or learnable fusion strategy. Their fused images have lower brightness and contrast values than those of the other methods do, which indicates a weaker ability to retain detailed texture information. In Figure 3a, the fused images of DenseFuse, NestFuse, and SwinFuse cannot display the text information of the pedestrian and the store billboard. In Figure 3b, the target on the building is hard to distinguish from the background. In Figure 3c,d, the tree branches lack details. The fused images of GANMcC retain the sharpness of the target in the infrared images. However, the background information, such as the plants and buildings, is fuzzy and lacks details. Res2Fusion offers better visual effects than the above methods do. In Figure 3a, the store billboard and pedestrian details are retained from the source images. Nevertheless, in Figure 3b, the target over the building is still not clear. In Figure 3c, the details of the tree branches are retained, but in Figure 3d, more specific details of the tree are lost. The fused images of IFCNN, SwinFusion, U2Fusion, and IFormerFusion generate fused images that can balance the gradient and intensity information. The targets are displayed with rich texture details and sharp edges. The details of background information, including the plants and buildings, are retained. However, IFormerFusion obtains the best visual effects and retains rich texture details and sharp edges.

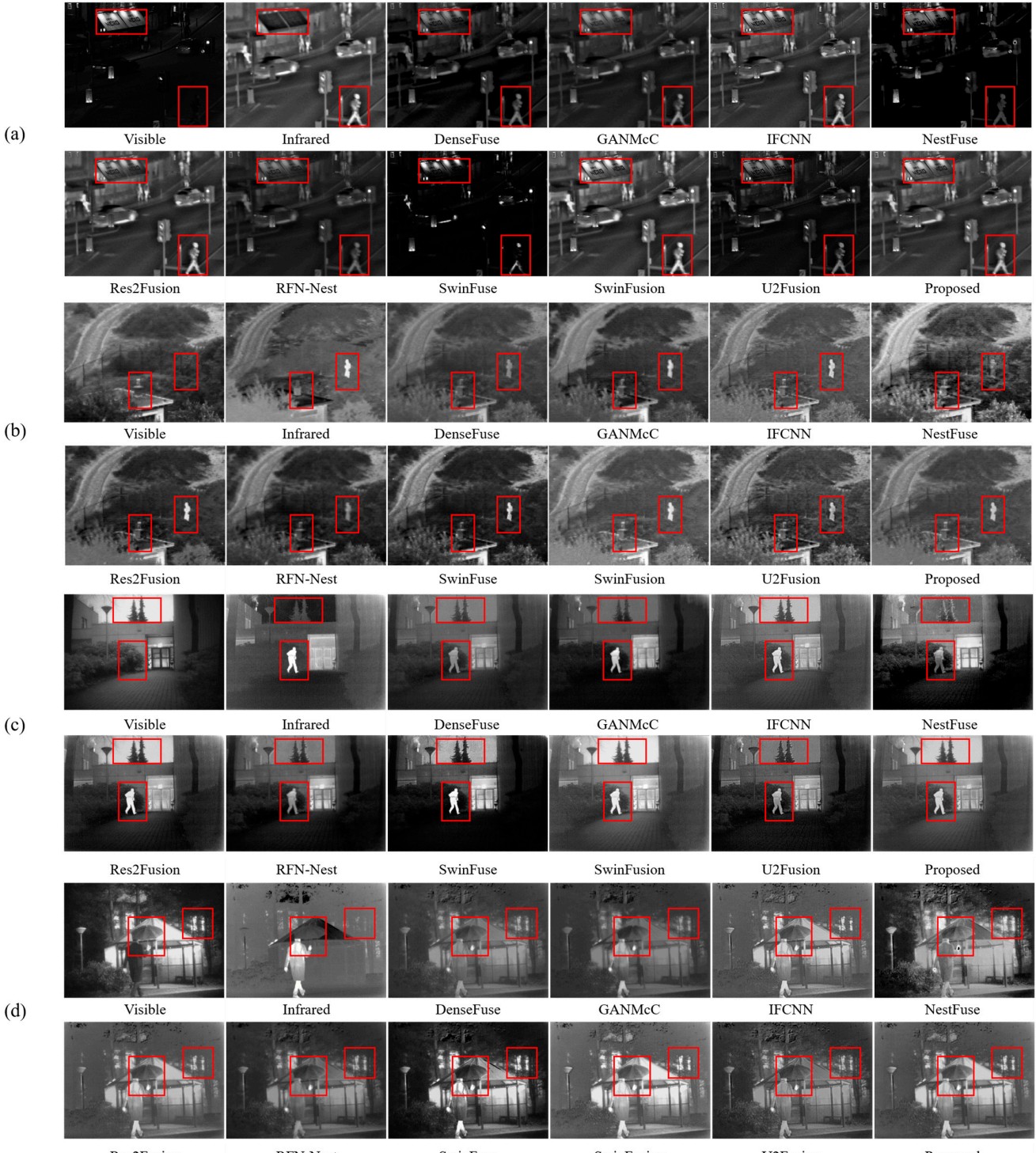

**Figure 3.** Subject comparisons of images selected from the TNO dataset: (**a**) street; (**b**) Nato camp; (**c**) Kaptein 1123; (**d**) Kaptein 1654. Some targets and details are highlighted with red boxes.

Quantitative Evaluation: To quantitatively evaluate IFormerFusion and the comparable methods, six metrics are calculated using 21 pairs of images. The result is shown in Figure 4. The average value of each metric is shown in Table 1. IFormerFusion obtains the best values in MI, FMI, PSNR, VIFF, and Qabf, and the second-best value in SSIM. More specifically, the highest MI and FMI values show that the proposed method can transfer more feature and edge information from the source images to the fused image. The highest PSNR value shows that the proposed method causes the least information distortion during

fusion. The highest VIFF value shows that the proposed method has more effective visual information. The highest $Q_{abf}$ value shows that the proposed method can obtain more visual information from the source images. The DenseFuse has the highest SSIM value, which indicates the advantage of structural information maintenance. However, DenseFuse is weaker in terms of the other metrics. Above all, the result indicates that IFormerFusion produces the best fusion results.

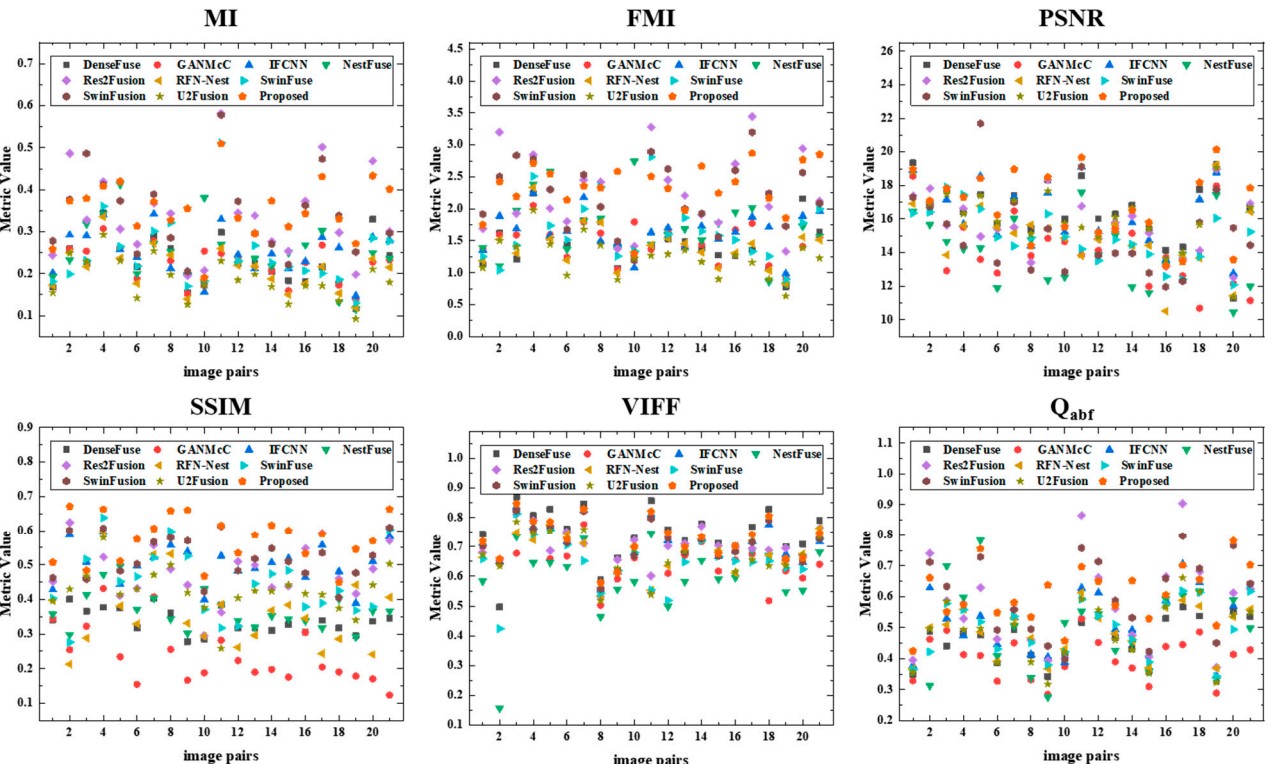

**Figure 4.** Object comparisons of 21 pairs of images selected from the TNO dataset.

**Table 1.** Average results of 21 pairs of images from the TNO dataset. The best values are in **bold**, and the second-best values are in *italic*.

| Method | MI | FMI | PSNR | SSIM | VIFF | $Q_{abf}$ |
|--------|------|------|------|------|------|------|
| DenseFuse | 1.4517 | 0.2261 | *16.5174* | **0.7459** | 0.4592 | 0.3383 |
| GANMcC | 1.4796 | 0.2244 | 14.2352 | 0.6424 | 0.4012 | 0.2380 |
| IFCNN | 1.6518 | 0.2510 | 16.1784 | 0.7126 | 0.5170 | 0.5041 |
| NestFuse | 1.6923 | 0.2524 | 13.9948 | 0.5967 | 0.4933 | 0.3667 |
| Res2Fusion | *2.2548* | 0.3380 | 15.6556 | 0.6998 | 0.5707 | 0.4764 |
| RFN-Nest | 1.4050 | 0.2084 | 15.0033 | 0.6776 | 0.4881 | 0.3598 |
| SwinFuse | 1.6524 | 0.2460 | 14.9131 | 0.6492 | 0.4960 | 0.4453 |
| SwinFusion | 2.2237 | *0.3389* | 15.1855 | 0.7102 | *0.5941* | *0.5216* |
| U2Fusion | 1.2451 | 0.1865 | 16.0628 | 0.6737 | 0.4809 | 0.4249 |
| Proposed | **2.3341** | **0.3538** | **16.7712** | *0.7258* | **0.6085** | **0.5765** |

### 4.4. Results on the OSU Dataset

Visual Quality Evaluation: A pair of images are selected to evaluate the visual effects of IFormerFusion and nine comparable methods. The results are shown in Figure 5. In the visible image, the pedestrian in the shadow of the building is hard to distinguish, while they are clearly observable in the infrared image. In the fused images of DenseFuse, GANMcC, NestFuse, and RFN-Nest, the gradient of the pedestrian is close to the building. Moreover, the edge of the pedestrian is blurred by the background building in GANMcC and RFN-Nest. In the fused images of IFCNN, Res2Fusion, SwinFuse, SwinFusion, and

IFormerFusion, the brightness of the pedestrian is similar to that of the infrared images, which indicates that the information in the infrared images is well retained. In the fused images of GANMcC and RFN-Nest, the sculpture on the lawn is blurred. The constrain of the lawn in the fused images of IFCNN, NestFuse, SwinFuse, and U2Fusion is discordant, which is more influenced by the infrared images. Thus, the texture detail of the lawn is lost. The lawn in the fused images of Res2Fusion, SwinFusion, and IFormerFusion has better visual effects. Above all, the proposed method, IFormerFusion, can retain texture detail and fuse with a balance of the gradient and intensity.

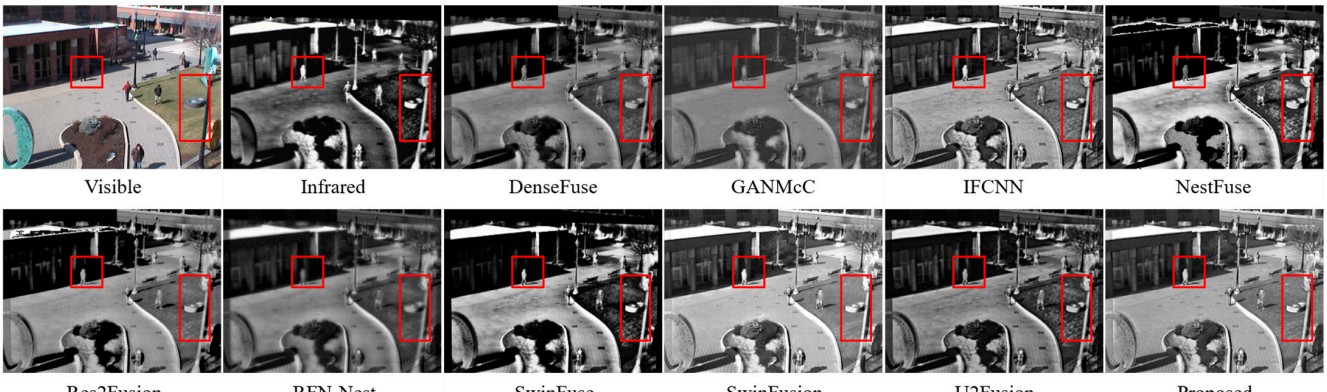

**Figure 5.** Subject comparisons of images selected from the OSU dataset. Some targets and details are highlighted with red boxes.

Quantitative Evaluation: To quantitatively evaluate IFormerFusion and compare the methods, six metrics are calculated using 40 pairs of images. The result is shown in Figure 6. The average value of each metric is shown in Table 2. IFormerFusion obtains the best results in terms of PSNR, VIFF, and Qabf, which indicates that the proposed method has the least information distortion and obtains the best visual effects during fusion. The proposed method lags behind the best method by a narrow margin in terms of MI, FMI, and SSIM. Thus, the result indicates that IFormerFusion can transfer a lot of information from the source images and produces the best results.

**Table 2.** Average results of 40 pairs of images from the OSU dataset. The best values are in **bold**, and the second-best values are in *italic*.

| Method | MI | FMI | PSNR | SSIM | VIFF | $Q_{abf}$ |
|---|---|---|---|---|---|---|
| DenseFuse | 1.8942 | 0.2634 | 14.7407 | 0.5988 | 0.3271 | 0.3906 |
| GANMcC | 1.8014 | 0.2548 | 14.6034 | 0.5377 | 0.2943 | 0.2120 |
| IFCNN | 1.9484 | 0.2602 | 14.7524 | 0.6074 | 0.3387 | *0.5322* |
| NestFuse | 2.3474 | 0.3280 | 12.4301 | 0.4739 | 0.3113 | 0.3910 |
| Res2Fusion | 2.6372 | 0.3557 | *15.0758* | **0.6268** | 0.3614 | 0.5013 |
| RFN-Nest | 1.8255 | 0.2496 | 14.6995 | 0.5699 | 0.3163 | 0.2588 |
| SwinFuse | 2.3002 | 0.3121 | 13.0634 | 0.5560 | 0.3294 | 0.4224 |
| SwinFusion | **2.7763** | **0.3693** | 13.9279 | 0.6125 | *0.3772* | 0.5296 |
| U2Fusion | 1.7818 | 0.2430 | 14.7607 | 0.6074 | 0.3300 | 0.4627 |
| Proposed | *2.7155* | *0.3636* | **15.2345** | *0.6137* | **0.4014** | **0.5865** |

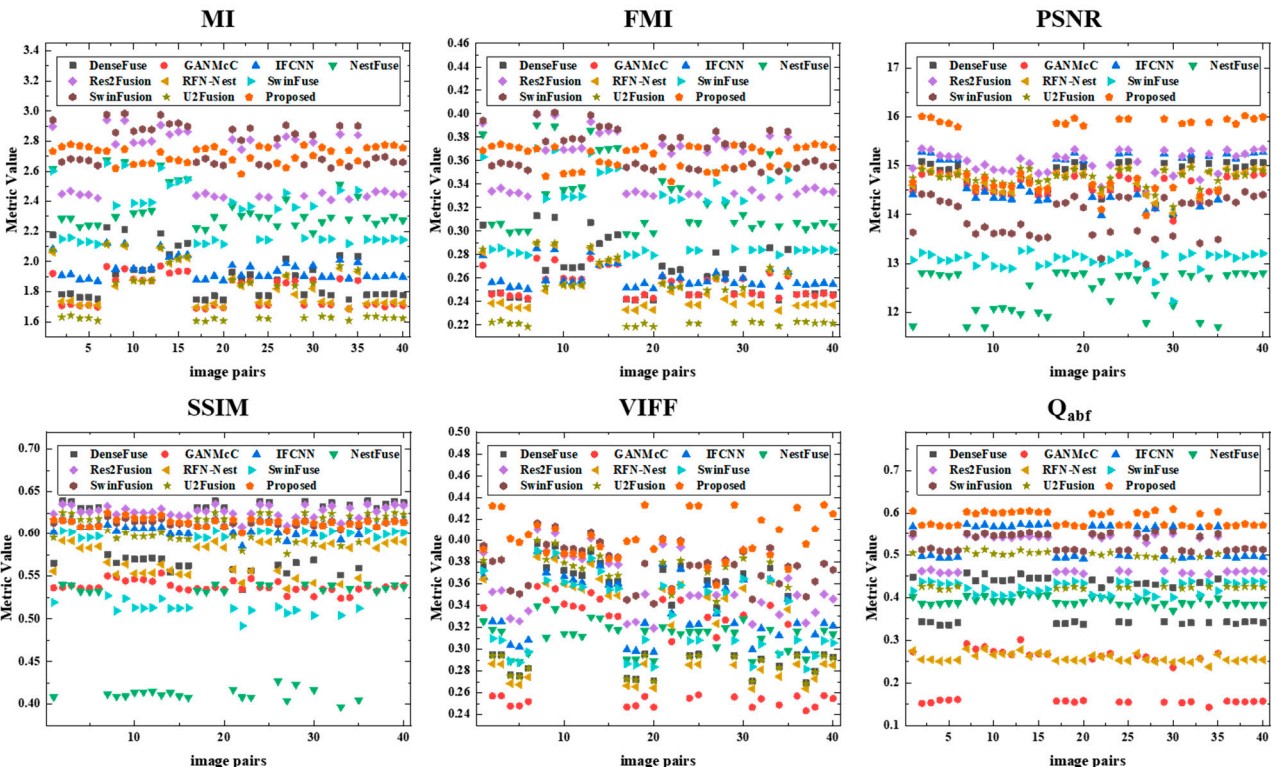

**Figure 6.** Object comparisons of 40 pairs of images selected from the OSU dataset.

### 4.5. Results on the Road Scene Dataset

Visual Quality Evaluation: A pair of images is selected to evaluate the visual effects of IFormerFusion and nine comparable methods. The result is shown in Figure 7. In the fused images of DenseFuse, GANMcC, Res2Fusion, and RFN-Nest, the tree on the left is hard to distinguish from the background. On the right of the images, the texture of the tire rim is rich, and the people standing by the area are fuzzy. In the fused image of NestFuse, though the tree branches are clear, artifacts exist in the car light, which indicates that NestFuse cannot balance the light information. In the fused images of IFCNN, SwinFuse, SwinFusion, U2Fusion, and IFormerFusion, the details of tree branches can be distinguished on the left. On the right, the tire rim is sharp, and the people in front of the car can be distinguished.

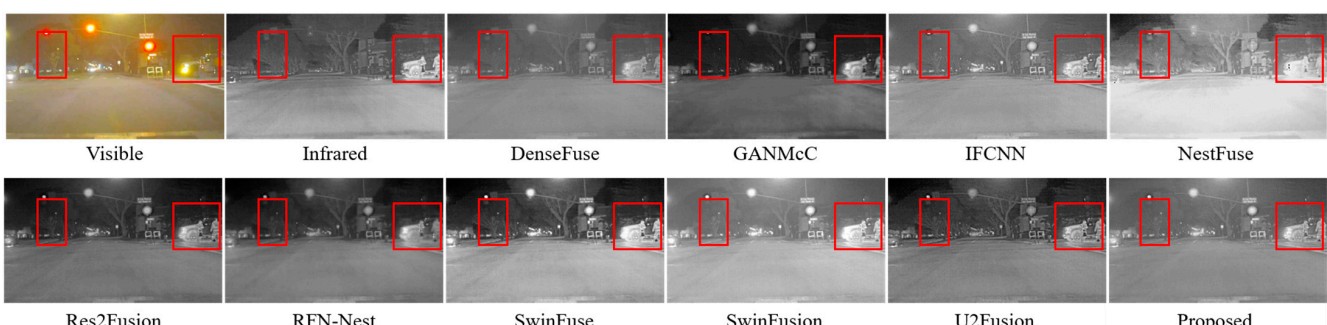

**Figure 7.** Subject comparisons of images selected from the Road Scene dataset. Some targets and details are highlighted with red boxes.

Quantitative Evaluation: To quantitatively evaluate IFormerFusion and compare methods, six metrics are calculated on 221 pairs of images from Road Scene datasets. The average value of each metric is shown in Table 3. The NestFuse obtains the best results in terms of MI and FMI. However, as shown in Figure 7, artifacts exist on the car, which

will introduce extra information to the fused images to increase the MI and FMI values. IFormerFusion obtains the best values in terms of PSNR, VIFF, and $Q_{abf}$. IFormerFusion obtains the second-best value for SSIM, behind that of DenseFuse. DenseFuse also obtains second-best result in terms of PNSR. However, DenseFuse produces unsatisfactory results in other metrics; in other words, DenseFuse can retain the structure and information from the source images, but has worse visual effects and less information. Above all, the result demonstrates that IFormerFusion produces the best fusion results.

**Table 3.** Average results of 221 pairs of images from the Road Scene dataset. The best values are in **bold**, and the second-best values are in *italic*.

| Method | MI | FMI | PSNR | SSIM | VIFF | $Q_{abf}$ |
|--------|-----|-----|------|------|------|------|
| DenseFuse | 2.0284 | 0.2874 | *16.4269* | **0.7230** | 0.4263 | 0.3827 |
| GANMcC | 1.9118 | 0.2645 | 13.3721 | 0.6220 | 0.3737 | 0.3360 |
| IFCNN | 2.0323 | 0.2827 | 16.3297 | 0.6860 | 0.4570 | *0.5449* |
| NestFuse | **2.6735** | **0.3685** | 11.9909 | 0.5784 | 0.3880 | 0.3754 |
| Res2Fusion | *2.4273* | 0.3338 | 14.1526 | 0.6634 | *0.4782* | 0.5135 |
| RFN-Nest | 1.9242 | 0.2640 | 13.9858 | 0.6344 | 0.4048 | 0.3129 |
| SwinFuse | 2.2021 | 0.2977 | 14.3088 | 0.6566 | 0.4464 | 0.5066 |
| SwinFusion | 2.3399 | *0.3339* | 14.3191 | 0.6932 | 0.4660 | 0.4611 |
| U2Fusion | 1.9364 | 0.2654 | 15.7728 | 0.6701 | 0.4409 | 0.5221 |
| Proposed | 2.2308 | 0.3158 | **17.0110** | *0.7052* | **0.4818** | **0.5490** |

*4.6. Computation Efficiency*

Moreover, we provide the computation efficiency of IFormerFusion and comparable methods, as shown in Table 4. DenseFuse, IFCNN, and NestFuse have a high computation efficiency because these methods consist of convolutional layers and simple fusion strategies. The RFN-Nest and SwinFuse one have a low computational efficiency because these methods utilize attention-based fusion strategies. Though GANMcC and U2Fusion also consist of convolutional layers, the computation is more complex. Thus, these methods have a lower computational efficiency. Res2Fusion and SwinFusion have the lowest computational efficiency because more complex attention mechanisms are used in these methods. However, the proposed IFormerFusion has a competitive computational efficiency and retains the linear computational complexity of the image size. All the methods are tested using a public code.

**Table 4.** Computation efficiency of IFomrerFusion and compared methods (unit: second). The best values are in **bold**, and the second-best values are in *italic*.

| Method | TNO | OSU | RoadScene |
|--------|-----|-----|-----------|
| DenseFuse | **0.0060** | **0.0079** | **0.0060** |
| GANMcC | 1.8520 | 1.0068 | 0.5242 |
| IFCNN | *0.0160* | *0.0132* | *0.0119* |
| NestFuse | 0.0167 | 0.0213 | 0.0144 |
| Res2Fusion | 10.208 | 0.919 | 2.937 |
| RFN-Nest | 0.2687 | 0.0510 | 0.1752 |
| SwinFuse | 0.1229 | 0.1248 | 0.1929 |
| SwinFusion | 8.6211 | 2.2020 | 5.1209 |
| U2Fusion | 2.3107 | 1.2639 | 0.6492 |
| Proposed | 0.3479 | 0.2678 | 0.2488 |

## 5. Conclusions

We propose IFormerFusion, a cross-domain frequency information learning infrared and visible image fusion network based on the Inception Transformer. The IFormer mixer based on the Inception Transformer consists of the high-frequency mixer, which consists of max pooling and convolution paths, and the low-frequency mixer, which contains a

criss-cross attention path. The high-frequency mixer can take the advantage of convolution and max-pooling for capturing high-frequency information. The low-frequency mixer can establish long-range dependency, and the criss-cross attention can reduce the computational complexity. Thus, the IFormer mixer can learn information in a wide frequency range. Moreover, the high-frequency information is traded in the cross-domain frequency fusion part to achieve the sufficient integration of complementary features and strengthen the capability to retain texture. The proposed IFormerFusion can comprehensively learn features from high- and low-frequency information to retain texture details and maintain the structure.

We conducted experiments on TNO, OSU, and Road Scene datasets and compared them with nine advanced deep-learning methods using six metrics. The results demonstrate that IFormerFusion performs well at preserving the structure and retaining texture details. In addition, IFormerFusion presents the balanced intensity of the targets and background in the fused image. IFormerFusion also has a competitive computational efficiency.

**Author Contributions:** Conceptualization, Z.X.; methodology, Z.X.; software, Z.X.; validation, Z.X.; formal analysis, Z.X.; investigation, Q.H.; resources, H.H., Q.H. and X.Z.; data curation, H.H.; writing—original draft preparation, Z.X.; writing—review and editing, X.Z.; visualization, Z.X. All authors have read and agreed to the published version of the manuscript.

**Funding:** This research received no external funding.

**Data Availability Statement:** Test images from TNO and OSU datasets are available at https://github.com/xiongzhangzzz/IFormerFusion (accessed on 21 February 2023). The Road Scene dataset can download from https://github.com/StaRainJ/road-scene-infrared-visible-images (accessed on 3 December 2022). The MSRS dataset can download from https://github.com/Linfeng-Tang/MSRS (accessed on 3 December 2022). The code and model are publicly accessible at https://github.com/xiongzhangzzz/IFormerFusion (accessed on 21 February 2023).

**Acknowledgments:** Thanks to the author of the compared methods for providing public codes and the author of the MSRS, TNO, OSU and Road Scene datasets for providing public datasets.

**Conflicts of Interest:** The authors declare no conflict of interest.

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
