# Peer review of "IFormerFusion: Cross-Domain Frequency Information Learning for Infrared and Visible Image Fusion Based on the Inception Transformer"

_remotesensing, doi:10.3390/rs15051352_

Round 1
Reviewer 1 Report
Please see these minor comments below.
Page 1 line 32: space after citation [1] and for the rest of he paper
Pg1 line 40: get good sounds awkward
Page 2 line 44: four types: then use a colon
Page 3 line 137 again colon
Page 6 line 218: what 8 advanced fusion methods? Okay you get to this on 223, put it on both places please
Page 8 lines 238-257: this is a pretty qualitative analysis, human eyes tend to lack the dynamic range of computer algorithms so I tend to differ to more quantitative metrics, but this is fine to describe quantitative results, just make it explicit these are qualitative and describing the subsequent paragraph
Overall a sound paper with a mostly well outlined methodology, please see comments above.
Here are some articles by my colleagues and I using thermal infrared and visible light imaging and deep learning with CNNs to automate the detection of explosive ordnances that have appeared in The Leading Edge, Journal of Conventional Weapons Destruction, and Remote Sensing.
Cheers,
Tim de Smet
Jebens, Martin and White, Rob (2021) "Remote Sensing and Artificial Intelligence in the Mine Action Sector," The Journal of Conventional Weapons Destruction: Vol. 25 : Iss. 1 , Article 28. Available at: https://commons.lib.jmu.edu/cisr-journal/vol25/iss1/28
Jebens, Martin; Sawada, Ph.D., Hideyuki; Shen, Junjie; and Tollefsen, Erik (2020) "To What Extent Could the Development of an Airborne Thermal Imaging Detection System Contribute to Enhance Detection?," The Journal of Conventional Weapons Destruction: Vol. 24 : Iss. 1 , Article 14. Available at: https://commons.lib.jmu.edu/cisr-journal/vol24/iss1/14
Fardoulis, John; Depreytere, Xavier; Gallien, Pierre; Djouhri, Kheria; Abdourhmane, Ba; and Sauvage, Emmanuel (2020) "Proof: How Small Drones Can Find Buried Landmines in the Desert Using Airborne IR Thermography," The Journal of Conventional Weapons Destruction: Vol. 24 : Iss. 2 , Article 15. Available at: https://commons.lib.jmu.edu/cisr-journal/vol24/iss2/15
Baur, J.; Steinberg, G.; Nikulin, A.; Chiu, K.; de Smet, T.S. Applying Deep Learning to Automate UAV-Based Detection of Scatterable Landmines. Remote Sens. 2020, 12, 859. https://doi.org/10.3390/rs12050859
Author Response
Dear Reviewers,
Thanks for your effort and time to review this manuscript. The authors appreciate all your comments and suggestions. We uploaded the file of the revised manuscript. Each comment will be directly addressed regarding the modified manuscript by the track changes mode in MS Word.
Comments and Suggestions for Authors
Please see these minor comments below.
- Page 1 line 32: space after citation [1] and for the rest of the paper
Response for comment 1:
Thanks for your kind suggestion. We apologize for the format problem and add space after each citation.
- Pg1 line 40: get good sounds awkward
Response for comment 2:
Thanks for your kind suggestion. We have revised the sentence. Moreover, we have asked for extensive English editing services from MDPI to improve this manuscript.
- Page 2 line 44: four types: then use a colon
Response for comment 3:
Thanks for your kind suggestion. We apologize for the format problem and change the comma to the colon.
- Page 3 line 137 again colon
Response for comment 4:
Thanks for your kind suggestion. We apologize for the format problem and we use a colon in all sentences in this manuscript while categorizing.
- Page 6 line 218: what 8 advanced fusion methods? Okay you get to this on 223, put it on both places please
Response for comment 5:
Thanks for your kind suggestion. We explain the name of the method in both places.
- Page 8 lines 238-257: this is a pretty qualitative analysis, human eyes tend to lack the dynamic range of computer algorithms so I tend to differ to more quantitative metrics, but this is fine to describe quantitative results, just make it explicit these are qualitative and describing the subsequent paragraph
Response for comment 6:
Thanks for your kind suggestion. We add “Visual Quality Evaluation” and “Quantitatively Evaluation” at the beginning of the paragraph to distinguish between qualitative analysis and quantitative analysis.
More specific analyses about quantitative metrics are applied in “Quantitatively Evaluation”, such as “Quantitatively Evaluation: To quantitatively evaluate IFormerFusion and compare methods, six metrics are calculated on 21 pairs of images. The result shows in Figure 4. The average value of each metric is shown in Table 1. The best values are in bold, and the second-best values are in italic in all tables in this paper. IFormerFusion obtains the best values in MI, FMI, PSNR, VIFF, and Qabf, and the second-best value in SSIM More specifically, the highest MI and FMI show that the proposed method can transfer more feature and edge information from source images to the fused image. The highest PSNR shows that the proposed method has the least information distortion while fusion. The highest VIFF shows that the proposed method has more effective visual information. The highest Qabf shows that the proposed method can obtain more visual information from the source images. The DenseFuse has the highest SSIM, which indicates the advantage of structural information maintenance. However, DenseFuse is weaker in other metrics. Above all, the result indicates that IFormerFusion performs the best fusion results.”
Overall a sound paper with a mostly well outlined methodology, please see comments above.
- Here are some articles by my colleagues and I using thermal infrared and visible light imaging and deep learning with CNNs to automate the detection of explosive ordnances that have appeared in The Leading Edge, Journal of Conventional Weapons Destruction, and Remote Sensing.
Jebens, Martin and White, Rob (2021) "Remote Sensing and Artificial Intelligence in the Mine Action Sector," The Journal of Conventional Weapons Destruction: Vol. 25 : Iss. 1 , Article 28. Available at: https://commons.lib.jmu.edu/cisr-journal/vol25/iss1/28
Jebens, Martin; Sawada, Ph.D., Hideyuki; Shen, Junjie; and Tollefsen, Erik (2020) "To What Extent Could the Development of an Airborne Thermal Imaging Detection System Contribute to Enhance Detection?," The Journal of Conventional Weapons Destruction: Vol. 24 : Iss. 1 , Article 14. Available at: https://commons.lib.jmu.edu/cisr-journal/vol24/iss1/14
Fardoulis, John; Depreytere, Xavier; Gallien, Pierre; Djouhri, Kheria; Abdourhmane, Ba; and Sauvage, Emmanuel (2020) "Proof: How Small Drones Can Find Buried Landmines in the Desert Using Airborne IR Thermography," The Journal of Conventional Weapons Destruction: Vol. 24 : Iss. 2 , Article 15. Available at: https://commons.lib.jmu.edu/cisr-journal/vol24/iss2/15
Baur, J.; Steinberg, G.; Nikulin, A.; Chiu, K.; de Smet, T.S. Applying Deep Learning to Automate UAV-Based Detection of Scatterable Landmines. Remote Sens. 2020, 12, 859. https://doi.org/10.3390/rs12050859
Response for comment 7:
Thanks for your kind suggestion. We have read these excellent articles carefully. We have cited these articles as reference [25], [26], [27], [28].
- Jebens, M.; White, R. Remote Sensing and Artificial Intelligence in the Mine Action Sector. The Journal of Conventional Weapons Destruction 2021, Vol. 25 : Iss. 1, Article 28.
- Jebens, M.; Sawada, H. To What Extent Could the Development of an Airborne Thermal Imaging Detection System Contribute to Enhance Detection? The Journal of Conventional Weapons Destruction 2020, Vol. 24 : Iss. 1, Article 14.
- Fardoulis, J.; Depreytere, X.; Gallien, P.; Djouhri, K.; Abdourhmane, B.; Sauvage, E. Proof: How Small Drones Can Find Buried Landmines in the Desert Using Airborne IR Thermography. The Journal of Conventional Weapons Destruction 2020, Vol. 24 : Iss. 2, Article 15.
- Baur, J.; Steinberg, G.; Nikulin, A.; Chiu, K.; de Smet, T.S. Applying Deep Learning to Automate UAV-Based Detection of Scatterable Landmines. Remote Sensing 2020, 12, doi:10.3390/rs12050859.
- Dosovitskiy, A.; Beyer, L.; Kolesnikov, A.; Weissenborn, D.; Zhai, X.; Unterthiner, T.; Dehghani, M.; Minderer, M.; Heigold, G.; Gelly, S.; et al. An Image Is Worth 16x16 Words: Transformers for Image Recognition at Scale. ICLR 2021.

Reviewer 2 Report
This manuscript presents a cross-domain frequency information learning for infrared and visible image fusion. The main contribution of this work is that the inception transformer is applied for infrared and visible image fusion. Experiments on several datasets demonstrate the effectiveness of the proposed method. However, there are still several problems need to be further solved before possible publication.
1) Please clarify the advantage of the inception transformer compared to CNN models. Why do you use the inception transformer?
2) Please highlight the motivation of this manuscript. What are the key contribution of this paper?
3) All vectors and matrices should be in bold.
4) For Fig. 3, the local regions of different results are not consistent. Please revise them. Furthermore, it can be seen that the compared method is superior than the proposed method. Please provide detailed reason.
5) Several related publications should be cited and reviewed, such as "A novel infrared and visible image fusion algorithm based on shift-invariant dual-tree complex shearlet transform and sparse representation".
Author Response
Dear Reviewers,
Thanks for your effort and time to review this manuscript. The authors appreciate all your comments and suggestions. We uploaded the file of the revised manuscript. Each comment will be directly addressed regarding the modified manuscript by the track changes mode in MS Word.
Comments and Suggestions for Authors
This manuscript presents a cross-domain frequency information learning for infrared and visible image fusion. The main contribution of this work is that the inception transformer is applied for infrared and visible image fusion. Experiments on several datasets demonstrate the effectiveness of the proposed method. However, there are still several problems need to be further solved before possible publication.
1) Please clarify the advantage of the inception transformer compared to CNN models. Why do you use the inception transformer?
Response for comment 1:
Thanks for your kind suggestion. We have tried our best to clarify the advantage of the inception transformer compared to CNN models. We use the inception transformer because the Inception Transformer has greater efficiency through a channel splitting mechanism to adopt parallel convolution/max-pooling paths and self-attention path to learn information within a wide frequency range, which is added to Section 2.1 Vision Transformer.
2) Please highlight the motivation of this manuscript. What are the key contribution of this paper?
Response for comment 2:
Thanks for your kind suggestion. We have revised the second and third paragraphs of 1. Introduction to explain the motivation of this manuscript and the key contribution of this paper. The changes are as follows:
However, there are still some drawbacks. Firstly, the existing methods fail to learn information from a wide frequency range, which is important for infrared and visible image fusion tasks. Secondly, the existing methods lack consideration about the relationship between the frequency information of the source images.
Above all, we proposed IFormerFusion based on the Inception Transformer and cross-domain frequency fusion. On the one hand, we design the IFormer mixer based on the Inception Transformer, which adopts convolution/ max-pooling paths to process high-frequency information and a criss-cross attention path to process low-frequency information. The IFormer mixer is designed to capture the high-frequency information to Transformers structure and learn information from a wide frequency range. On the other hand, cross-domain frequency fusion can trade high-frequency information across the source images to guide the model to retain more high-frequency information. IFormerFusion has four parts: feature extraction, cross-domain frequency fusion, feature reconstruction, and fused image reconstruction. The feature extractor can effectively extract comprehensive features from a wide frequency range. The cross-domain frequency fusion can learn features in a wide frequency range and trade high-frequency information across the source images to strengthen the texture-retaining capability of the model. The fusion results are concatenated and fed to the Inception Transformer-based feature reconstruction part to reconstruct deep features. Finally, a CNN-based fused image reconstruction part is utilized to reconstruct images. Above all, the main contributions of this work can be summarized as follows:
- We propose an infrared and visible image fusion method, IFormerFusion, which can efficiently learn features from source images in a wide frequency range. IFormerfusion can sufficiently retain texture detail and maintain the structure of source images.
- We design the IFormer mixer that consists of convolution/ max-pooling paths and a criss-cross attention path. The convolution/ max-pooling can learn high-frequency information, while the criss-cross attention path can learn low-frequency information.
- The cross-domain frequency fusion can trade high-frequency information across the source images to sufficiently learn comprehensive features and strengthen the capability of texture retaining.
- Experiments conducted using the TNO, OSU, and Road Scene datasets show that IFormerFusion obtains better results in both visual quality evaluation and quantitatively evaluation.
3) All vectors and matrices should be in bold.
Response for comment 3:
Thanks for your kind suggestion. We apologize for the format problem and make all vectors and matrices bold.
4) For Fig. 3, the local regions of different results are not consistent. Please revise them. Furthermore, it can be seen that the compared method is superior than the proposed method. Please provide detailed reason.
Response for comment 4:
Thanks for your kind suggestion. We have adjusted the local regions of different results in Figure.3 . And we add “Visual Quality Evaluation” and “Quantitatively Evaluation” at the beginning of the paragraph to distinguish between qualitative analysis and quantitative analysis.
More specific analysis about quantitative metrics are applied in “Quantitatively Evaluation” to explain why the compared method is superior than the proposed method in some metric. such as “Quantitatively Evaluation: To quantitatively evaluate IFormerFusion and com-pared methods, six metrics were are calculated on 21 pairs of images. The result shows in Figure 4. The average value of each metric is shown in Table 1. The best values are in bold, and the second-best values are in italic in all tables in this paper. IFormerFusion gets obtains the best values in MI, FMI, PSNR, VIFF, and Qabf and, which indicates that IFormerFusion can fuse images with high quality and visual effects. IFormerFu-sion gets the second-best value in SSIM, which indicates that IFormerFusion has well structure maintenance. The result indicates that IFormerFusion performs the best fu-sion results. More specifically, the highest MI and FMI show that the proposed method can transfer more feature and edge information from source images to the fused image. The highest PSNR shows that the proposed method has the least information distortion while fusion. The highest VIFF shows that the proposed method has more effective visual information. The highest Qabf shows that the proposed method can obtain more visual information from the source images. The DenseFuse has the highest SSIM, which indicates the advantage of structural information maintenance. Howerver, DenseFuse is weaker in other metrics. Above all, the result indicates that IFormerFu-sion performs the best fusion results.” Similar analysis on other datasets is also changed.
5) Several related publications should be cited and reviewed, such as "A novel infrared and visible image fusion algorithm based on shift-invariant dual-tree complex shearlet transform and sparse representation".
Response for comment 5:
Thanks for your kind suggestion. We have read these excellent articles carefully. We have cited these articles as reference [8] and [9]
- Yin, M.; Duan, P.; Liu, W.; Liang, X. A Novel Infrared and Visible Image Fusion Algorithm Based on Shift-Invariant Dual-Tree Complex Shearlet Transform and Sparse Representation. Neurocomputing 2017, 226, 182–191, doi:10.1016/j.neucom.2016.11.051.
- Yin, M.; Liu, W.; Zhao, X.; Yin, Y.; Guo, Y. A Novel Image Fusion Algorithm Based on Nonsubsampled Shearlet Transform. Optik 2014, 125, 2274–2282, doi:10.1016/j.ijleo.2013.10.064.

Reviewer 3 Report
The article is proposing novel method for fusion of infrared and visible images which can retain texture details and maintain structure of original image. The article is well written, but there are some concerns that should be addressed.
COnclusion section - Please add some more results and description in conclusion - strengthen it a little.
Line 88 - Reference. No year no connection to reference. I see that it is [17] in next sentence but you should at least add year.
Lines 91, 100, 102, 108, 121, 123, 124 "Ma et. al." - same as previous reference - add year
Line 149 - please rewrite it is little incomprehensible
Line 202 "SSIM" - add full name - structural similarity
Table 1, 2 and 3 - I would add information that it is average result in table description
Table 4 - Please bold the best result as in previous three tables.
It would be very interesting to determine and see the ratio of average values of SSIM, MI PSNR etc. (Tables 1-3) with computation efficiency (Table 4). In order to determine best model for certain amount of time.
Author Response
Dear Reviewers,
Thanks for your effort and time to review this manuscript. The authors appreciate all your comments and suggestions. We uploaded the file of the revised manuscript. Each comment will be directly addressed regarding the modified manuscript by the track changes mode in MS Word.
Comments and Suggestions for Authors
The article is proposing novel method for fusion of infrared and visible images which can retain texture details and maintain structure of original image. The article is well written, but there are some concerns that should be addressed.
Conclusion section - Please add some more results and description in conclusion - strengthen it a little.
Response for comment 1:
Thanks for your kind suggestion. We have added more results and descriptions in Conclusion section to strengthen it. The changes are as follows:
We propose IFormerFusion, a cross-domain frequency information learning infrared and visible image fusion network based on the Inception Transformer. IFormerFusion has four parts: feature extraction, cross-domain frequency fusion, feature reconstruction, and fused image reconstruction. We designed the IFormer mixer to learn features from a wide frequency range. The IFormer mixer based on the Inception Transformer consists of the high-frequency mixer, which consists of max pooling and convolution paths, and the low-frequency mixer, which contains a criss-cross attention path. The high-frequency mixer can take the advantage of convolution and max-pooling for capturing high-frequency information. The low-frequency mixer can establish long-range dependency and the criss-cross attention can reduce computational complexity. Thus, the IFormer mixer can learn information from a wide frequency range. Moreover, the high-frequency information trades off in the cross-domain frequency fusion part to achieve sufficient integration of complementary features and strengthen the capability of texture retaining. The proposed IFormerFusion can comprehensively learn features from high- and low-frequency information to retain texture detail and maintain the structure.
Line 88 - Reference. No year no connection to reference. I see that it is [17] in next sentence but you should at least add year.
Response for comment 2:
Thanks for your kind suggestion. We have revised this sentence to eliminate the ambiguity and added the year information.
Lines 91, 100, 102, 108, 121, 123, 124 "Ma et. al." - same as previous reference - add year.
Response for comment 3:
Thanks for your kind suggestion. We have added the year information to these citations.
Line 149 - please rewrite it is little incomprehensible
Response for comment 4:
Thanks for your kind suggestion. We have revised this paragraph and the former paragraph to avoid incomprehensible descriptions. We also add legends for Figure 1 to explain more precisely. The mistake might be because we missed the phrase, high-frequency information. We split the embedding result into three partitions including two high-frequency partitions and a low-frequency partition through the channel dimension and feed them into the IFormer mixer. Ï•h1 and Ï•h2 represent high-frequency information, Ï•l represents low-frequency information.
Line 202 "SSIM" - add full name - structural similarity
Response for comment 5:
Thanks for your kind suggestion. We have added full name structural similarity for SSIM.
Table 1, 2 and 3 - I would add information that it is average result in table description
Response for comment 6:
The author would like to thank the reviewer for the suggestion for adding information about the average result. Thus, we have added “average” to the table description. The changes are as follows:
Table 1. Average results on 21 pairs of images from the TNO dataset.
Table 2. Average results on 40 pairs of image from the OSU dataset.
Table 3. Average results on 221 pairs of images from the Road Scene dataset.
Table 4 - Please bold the best result as in previous three tables.
It would be very interesting to determine and see the ratio of average values of SSIM, MI PSNR etc. (Tables 1-3) with computation efficiency (Table 4). In order to determine best model for certain amount of time.
Response for comment 7:
Thanks for your kind suggestion. The best values are in bold, and the second-best values are in italic in Table 4.
Table 4. Computation efficiency of IFomrerFusion and compared methods (Unit: second)
Method |
TNO |
OSU |
RoadScene |
DenseFuse |
0.0060 |
0.0079 |
0.0060 |
GANMcC |
1.8520 |
1.0068 |
0.5242 |
IFCNN |
0.0160 |
0.0132 |
0.0119 |
NestFuse |
0.0167 |
0.0213 |
0.0144 |
Res2Fusion |
10.208 |
0.919 |
2.937 |
RFN-Nest |
0.2687 |
0.0510 |
0.1752 |
SwinFuse |
0.1229 |
0.1248 |
0.1929 |
SwinFusion |
8.6211 |
2.2020 |
5.1209 |
U2Fusion |
2.3107 |
1.2639 |
0.6492 |
Proposed |
0.3479 |
0.2678 |
0.2488 |
The author would like to thank the reviewer for the suggestion for calculating the ratio of average values of SSIM, MI PSNR etc. (Tables 1-3) with computation efficiency (Table 4) to determine best model for certain amount of time. We have calculated the ratios and the result is shown in Table 1, 2, 3. DenseFuse alwayls has the highest value. IFCNN and NestFuse usually have the second-highest value. It is because that due to the complexity of the model, the average time of each image generated by different algorithms has an exponential difference. However, the difference between image quality evaluation parameters is usually less than twice. The computation efficiency affects the ratio much more than evaluation metrics. If we rank all fusion method with the ratios of average values with computation efficiency, it seems to be similar with rank all fusion methods with computation efficiency. We have found proper way to show these tables in this manuscript.
Table 1. The ratios of average values with computation efficiency on 21 pairs of images from the TNO dataset.
Method |
MI |
FMI |
PSNR |
SSIM |
VIFF |
Qabf |
DenseFuse |
241.950 |
37.683 |
2752.900 |
124.317 |
76.533 |
56.383 |
GANMcC |
0.799 |
0.121 |
7.686 |
0.347 |
0.217 |
0.129 |
IFCNN |
103.238 |
15.688 |
1011.150 |
44.538 |
32.313 |
31.506 |
NestFuse |
101.335 |
15.114 |
838.012 |
35.731 |
29.539 |
21.958 |
Res2Fusion |
0.221 |
0.033 |
1.534 |
0.069 |
0.056 |
0.047 |
RFN-Nest |
5.229 |
0.776 |
55.837 |
2.522 |
1.817 |
1.339 |
SwinFuse |
13.445 |
2.002 |
121.343 |
5.282 |
4.036 |
3.623 |
SwinFusion |
0.258 |
0.039 |
1.761 |
0.082 |
0.069 |
0.061 |
U2Fusion |
0.539 |
0.081 |
6.951 |
0.292 |
0.208 |
0.184 |
Proposed |
6.709 |
1.017 |
48.207 |
2.086 |
1.749 |
1.657 |
Table 2. The ratios of average values with computation efficiency on 40 pairs of images from the OSU dataset.
Method |
MI |
FMI |
PSNR |
SSIM |
VIFF |
Qabf |
DenseFuse |
315.700 |
43.900 |
2456.783 |
99.800 |
54.517 |
65.100 |
GANMcC |
0.973 |
0.138 |
7.885 |
0.290 |
0.159 |
0.114 |
IFCNN |
121.775 |
16.263 |
922.025 |
37.963 |
21.169 |
33.263 |
NestFuse |
140.563 |
19.641 |
744.317 |
28.377 |
18.641 |
23.413 |
Res2Fusion |
0.258 |
0.035 |
1.477 |
0.061 |
0.035 |
0.049 |
RFN-Nest |
6.794 |
0.929 |
54.706 |
2.121 |
1.177 |
0.963 |
SwinFuse |
18.716 |
2.539 |
106.293 |
4.524 |
2.680 |
3.437 |
SwinFusion |
0.322 |
0.043 |
1.616 |
0.071 |
0.044 |
0.061 |
U2Fusion |
0.771 |
0.105 |
6.388 |
0.263 |
0.143 |
0.200 |
Proposed |
7.805 |
1.045 |
43.790 |
1.764 |
1.154 |
1.686 |
Table 3. The ratios of average values with computation efficiency on 221 pairs of images from the Road Scene dataset.
Method |
MI |
FMI |
PSNR |
SSIM |
VIFF |
Qabf |
DenseFuse |
338.067 |
47.900 |
2737.817 |
120.500 |
71.050 |
63.783 |
GANMcC |
1.032 |
0.143 |
7.220 |
0.336 |
0.202 |
0.181 |
IFCNN |
127.019 |
17.669 |
1020.606 |
42.875 |
28.563 |
34.056 |
NestFuse |
160.090 |
22.066 |
718.018 |
34.635 |
23.234 |
22.479 |
Res2Fusion |
0.238 |
0.033 |
1.386 |
0.065 |
0.047 |
0.050 |
RFN-Nest |
7.161 |
0.983 |
52.050 |
2.361 |
1.507 |
1.164 |
SwinFuse |
17.918 |
2.422 |
116.426 |
5.343 |
3.632 |
4.122 |
SwinFusion |
0.271 |
0.039 |
1.661 |
0.080 |
0.054 |
0.053 |
U2Fusion |
0.838 |
0.115 |
6.826 |
0.290 |
0.191 |
0.226 |
Proposed |
6.412 |
0.908 |
48.896 |
2.027 |
1.385 |
1.578 |

Reviewer 4 Report
The last part of the introduction can be improved.
a) Aiming at showing the advantages of the method, the authos describe the "conclusions" of the paper. Therefore, the conclusions are very weak, more like a summary wiothout the conclusion itself.
b) THe introduction section could be improved if a hypothesis is formulated. What problem will be solved based on what?
Method: line 134-...
this is not clear enough.
According to Fig 1, the input image is a set of C-matrixes (HxW). what are the base blocks? what defines N1? why is it necessary to increase the channel dimesnion (C)? why 60?
figure 1:
please review the symbols. I guess "H" is not tyhe same as the image size...
Please explain them. I did not understand N1, N2, N3, for example.
line 144:
Why to you split in this stage? How do you split the embedding image?
what do you mean by saying "three partitions'? How do you define the partitions? why 3? are they frequencies?
Do you mean H,L, and whatever in figure 1?
why does the second arrow habve no nme (such as H or L)?
what is H: high what frequencies? please, provide more details
In this paragraph you quote the Iformer mixer, but the real explanation of the Iformer mixer comes later. tehrefore, it is difficult to understandwhat happens in the process.
it is not cler here, why do I need an Iformer mixer?
Lines 144-145:
Why to you split in this stage? How do you split the embedding image? what do you mean by saying "three partitions'? How do you define the partitions? why 3? are they frequencies?
Do you mean H,L, and whatever in figure 1?
why does the second arrow have no name (such as H or L)?
what is H: high what frequencies?
please, provide more details. One can only understand this after reading section 3.2.
line 242:
strategy. The fused images of them (maybe Their fused images?)
line 261
Plese provide more information about the quality parameters. Using your text, one can hardly understand what are you measuring and evaluate why your method is better. MI, FMI, PSNR, VIFF, Qabf, SSIM, such terms should be explained. what do they measure and how? why is a high/low value better? This would improve the understanding of your tables.
figure 4:
data is not a series, tehrefore points should no be linked by lines. consider using another graphic.
conclusions:
in the introduction section yo provide more conclusions than here.
Author Response
Dear Reviewers,
Thanks for your effort and time to review this manuscript. The authors appreciate all your comments and suggestions. We uploaded the file of the revised manuscript. Each comment will be directly addressed regarding the modified manuscript by the track changes mode in MS Word.
Comments and Suggestions for Authors
The last part of the introduction can be improved.
- a) Aiming at showing the advantages of the method, the authors describe the "conclusions" of the paper. Therefore, the conclusions are very weak, more like a summary without the conclusion itself.
Response for comment a):
The author would like to thank the reviewer for the suggestion of improving Introduction and Conclusion sections.
- We have revised the last part of the Introduction section and removed some improper descriptions. The changes are as follows:
Above all, we proposed IFormerFusion based on the Inception Transformer and cross-domain frequency fusion. On the one hand, we design the IFormer mixer based on the Inception Transformer, which adopts convolution/ max-pooling paths to process high-frequency information and a criss-cross attention path to process low-frequency information. The IFormer mixer is designed to capture the high-frequency information to Transformers structure and learn information from a wide frequency range. On the other hand, cross-domain frequency fusion can trade high-frequency information across the source images to guide the model to retain more high-frequency information. IFormerFusion has four parts: feature extraction, cross-domain frequency fusion, feature reconstruction, and fused image reconstruction. The feature extractor can effectively extract comprehensive features from a wide frequency range. The cross-domain frequency fusion can learn features in a wide frequency range and trade high-frequency information across the source images to strengthen the texture-retaining capability of the model. The fusion results are concatenated and fed to the Inception Transformer-based feature reconstruction part to reconstruct deep features. Finally, a CNN-based fused image reconstruction part is utilized to reconstruct images.
- We also have revised the conclusion and added more results and descriptions to strengthen it. The changes are as follows:
We propose IFormerFusion, a cross-domain frequency information learning infrared and visible image fusion network based on the Inception Transformer. The IFormer mixer based on the Inception Transformer consists of the high-frequency mixer, which consists of max pooling and convolution paths, and the low-frequency mixer, which contains a criss-cross attention path. The high-frequency mixer can take the advantage of convolution and max-pooling for capturing high-frequency information. The low-frequency mixer can establish long-range dependency and the criss-cross attention can reduce computational complexity. Thus, the IFormer mixer can learn information from a wide frequency range. Moreover, the high-frequency information trades off in the cross-domain frequency fusion part to achieve sufficient integration of complementary features and strengthen the capability of texture retaining. The proposed IFormerFusion can comprehensively learn features from high- and low-frequency information to retain texture detail and maintain the structure.
- b) The introduction section could be improved if a hypothesis is formulated. What problem will be solved based on what?
Response for comment b):
The author would like to thank the reviewer for the suggestion of improving Introduction section. We have revised the last paragraph and added explanations of the problem to be solved. The changes are as follows:
However, there are still some drawbacks. Firstly, the existing methods fail to learn information from a wide frequency range, which is important for infrared and visible image fusion tasks. Secondly, the existing methods lack consideration about the relationship between the frequency information of the source images.
Above all, we proposed IFormerFusion based on the Inception Transformer and cross-domain frequency fusion. On the one hand, we design the IFormer mixer based on the Inception Transformer, which adopts convolution/ max-pooling paths to process high-frequency information and a criss-cross attention path to process low-frequency information. The IFormer mixer is designed to capture high-frequency information to Transformers structure and learn information from a wide frequency range. On the other hand, cross-domain frequency fusion can trade high-frequency information across the source images to guide the model to retain more high-frequency information. IFormerFusion has four parts: feature extraction, cross-domain frequency fusion, feature reconstruction, and fused image reconstruction. The feature extractor can effectively extract comprehensive features from a wide frequency range. The cross-domain frequency fusion can learn features in a wide frequency range and trade high-frequency information across the source images to strengthen the texture-retaining capability of the model. The fusion results are concatenated and fed to the Inception Transformer-based feature reconstruction part to reconstruct deep features. Finally, a CNN-based fused image reconstruction part is utilized to reconstruct images.
Method: line 134-...
this is not clear enough.
According to Fig 1, the input image is a set of C-matrixes (HxW). what are the base blocks? what defines N1?why is it necessary to increase the channel dimesnion (C)? why 60?
Response for comment:
Thanks for your kind suggestions. We apologize for not explaining the framework of the IFormerFusion well. We have revised the descriptions of the framework of the IFormerFusion to make it more clearly. The base block is referred to an Inception transformer block. N1 means the base block is repeated for N1 times. We increase the channel dimension to improve the model performance. Experiments in our manuscript are conducted on an NVIDIA GeForce RTX 3090. We set the channel dimension as 60 to make full use of the memory of the GPU according to the model design in the paper Inception Transformer (https://arxiv.org/pdf/2205.12956v2.pdf).
figure 1:
please review the symbols. I guess "H" is not the same as the image size...
Please explain them. I did not understand N1, N2, N3, for example.
Response for comment:
Thanks for your kind suggestions. We have revised the descriptions of the framework of the IFormerFusion and redrawn the figure 1 and 2.
- "H" is not the same as the image size. "H" represents the high-frequency channels fed into the high-frequency mixer. We have redrawn the figure to clarify it.
- N1, N2, N3 means the block in yellow box is repeated for N1, N2, N3
line 144:
Why do you split in this stage? How do you split the embedding image?
Response for comment:
Thanks for your kind suggestions. We have revised the paragraph (Page.6 Line.207) about how to split the embedding image.
- We split in this stage because we design the IFormer mixer to learn information from a wide frequency range. The high-frequency mixer can take the advantage of convolution and max-pooling for capturing high-frequency information. The low-frequency mixer can establish long-range dependency and the criss-cross attention can reduce computational complexity. We have added this to the last paragraph of 1. Introduction.
- We first increase the channel dimension. Then we split the image to high- and low-frequency parts through channel dimensions. The high-frequency part will be fed into a max-pooling path and a convolution path of the high mixer. The low-frequency part will be fed into the criss-cross attention path of the low mixer. With the input X , X can be divided into two parts through channel dimension, the high-frequency part and the low-frequency part , where Ch +Cl = C. Then, and are assigned to a high-frequency mixer and a low-frequency mixer respectively
what do you mean by saying "three partitions'? How do you define the partitions? why 3? are they frequencies?
Response for comment:
Thanks for your kind suggestions.
- With the input X , X can be divided into two parts through channel dimension, the high-frequency part and the low-frequency part , where Ch +Cl = C. Then, and are assigned to high-frequency mixer and low-frequency mixer respectively. In high-frequency mixer, the input is divided into and . is fed into the max pooling path. is fed into the parallel convolution path. In low-frequency mixer, is fed into the max pooling path. Thus, we have three patitions, , and . We have revised the paragraph (Page.6 Line.207) about how to split the embedding image.
- We have revised the paragraph (Page.4 Line.166) about three partitions. We split the embedding images into three parts, including two high-frequency partitions (H1 and H2) and a low-frequency partition(L), which is called partition in this manuscript.
- We have revised the paragraph (Page.4 Line.166) about how paths in high mixer and low mixer work. , and .are fed into different paths. The changes are as follows:
- Considering the sharp sensitiveness of the maximum filter and the detail perception of convolution operation, we propose two paths for high-frequency to take advantage of the sharpness sensitiveness of max-pooling and the detail perception capability of convolution layers and learn high-frequency information. (Page. 6 Line. 212)
- Considering the strong capability of the attention mechanism for learning global representation, we use criss-cross attention to establish long-range dependency to learn low-frequency information. (Page. 6 Line. 219)
Do you mean H,L, and whatever in figure 1?
why does the second arrow have no name (such as H or L)?
what is H: high what frequencies?please, provide more details
Response for comment:
Thanks for your kind suggestions. We have redrawn the figure. 1 to make it more clearly.
- H and L represent the high- and low-frequency information. We have changed it to H1, H2, and H3, which is corresponding to three paths of the IFormer mixer.
- We have added the name.
- "H" represents the high-frequency channels fed into the high-frequency mixer. We have added legends for figure. 1.
In this paragraph you quote the Iformer mixer, but the real explanation of the Iformer mixer comes later. therefore, it is difficult to understand what happens in the process.
it is not clear here, why do I need an Iformer mixer?
Response for comment:
Thanks for your kind suggestions. We have revised the paragraph about the descriptions of figure. 1. We have also redrawn the figure. 1 to make it more clearly. IFormer mixer is utilized to learn both high- and low-frequency information from source images.
Lines 144-145:
Why to you split in this stage? How do you split the embedding image? what do you mean by saying "three partitions'? How do you define the partitions? why 3? are they frequencies?
Do you mean H,L, and whatever in figure 1?
why does the second arrow have no name (such as H or L)?
what is H: high what frequencies?
please, provide more details. One can only understand this after reading section 3.2.
Response for comment:
Thanks for your kind suggestions. We have redrawn the figure. 1 and revised the descriptions to make it more clearly. These questions are the same as the former ones.
line 242:
strategy. The fused images of them (maybe Their fused images?)
Response for comment:
Thanks for your kind suggestion. We have revised the sentence. Moreover, we have asked for extensive English editing services from MDPI to improve this manuscript.
line 261
Please provide more information about the quality parameters. Using your text, one can hardly understand what are you measuring and evaluate why your method is better. MI, FMI, PSNR, VIFF, Qabf, SSIM, such terms should be explained. what do they measure and how? why is a high/low value better? This would improve the understanding of your tables.
Response for comment:
Thanks for your kind suggestion. We have added Section 4.1 Evaluation Metrics (Page. 7, Line 270) to explain the metrics used in this manuscript. We have revised the discussion part.
More specific analyses about quantitative metrics are applied in “Quantitatively Evaluation” to explain why the compared method is superior than the proposed method in some metric. such as “Quantitatively Evaluation: To quantitatively evaluate IFormerFusion and compare methods, six metrics are calculated on 21 pairs of images. The result shows in Figure 4. The average value of each metric is shown in Table 1. The best values are in bold, and the second-best values are in italic in all tables in this paper. IFormerFusion obtains the best values in MI, FMI, PSNR, VIFF, and Qabf, and the second-best value in SSIM More specifically, the highest MI and FMI show that the proposed method can transfer more feature and edge information from source images to the fused image. The highest PSNR shows that the proposed method has the least information distortion while fusion. The highest VIFF shows that the proposed method has more effective visual information. The highest Qabf shows that the proposed method can obtain more visual information from the source images. The DenseFuse has the highest SSIM, which indicates the advantage of structural information maintenance. However, DenseFuse is weaker in other metrics. Above all, the result indicates that IFormerFusion performs the best fusion results.”
figure 4:
data is not a series, therefore points should no be linked by lines. consider using another graphic.
Figure 4. Object comparisons of 21 pairs of images selected from the TNO dataset. |
Thanks for your kind suggestions. We hoped to visually demonstrate the relative size relationship of various methods of evaluating indicators. Thus, we follow the figure format in Ref. [14] and Ref. [42]. We have not fully considering that the data is not a series. Thus, we have removed the linking lines. Figure 4 and 6 is shown below.
Figure 6. Object comparisons of 40 pairs of images selected from the OSU dataset. |
Liu, X.; Gao, H.; Miao, Q.; Xi, Y.; Ai, Y.; Gao, D. MFST: Multi-Modal Feature Self-Adaptive Transformer for Infrared and Visible Image Fusion. Remote Sensing 2022, 14, doi:10.3390/rs14133233.- Wang, Z.; Chen, Y.; Shao, W.; Li, H.; Zhang, L. SwinFuse: A Residual Swin Transformer Fusion Network for Infrared and Visible Images. IEEE Transactions on Instrumentation and Measurement 2022, 71, 1–12, doi:10.1109/TIM.2022.3191664.

Round 2
Reviewer 2 Report
No more comments.
Reviewer 3 Report
The article is much better and I think it is ready for publication